# Mild proteasomal stress improves photosynthetic performance in Arabidopsis chloroplasts

Julia Grimmer[1], Stefan Helm[1], Dirk Dobritzsch[1], Gerd Hause[2], Gerta Shema[3], René P. Zahedi[3,4] &
Sacha Baginsky [1,5✉]

The proteasome is an essential protein-degradation machinery in eukaryotic cells that controls protein turnover and thereby the biogenesis and function of cell organelles. Chloroplasts import thousands of nuclear-encoded precursor proteins from the cytosol, suggesting that the bulk of plastid proteins is transiently exposed to the cytosolic proteasome complex. Therefore, there is a cytosolic equilibrium between chloroplast precursor protein import and proteasomal degradation. We show here that a shift in this equilibrium, induced by mild genetic proteasome impairment, results in elevated precursor protein abundance in the cytosol and significantly increased accumulation of functional photosynthetic complexes in protein import-deficient chloroplasts. Importantly, a proteasome *lid* mutant shows improved photosynthetic performance, even in the absence of an import defect, signifying that functional precursors are continuously degraded. Hence, turnover of plastid precursors in the cytosol represents a mechanism to constrain thylakoid membrane assembly and photosynthetic electron transport.

[1] Institute of Biochemistry and Biotechnology, Martin-Luther-University Halle-Wittenberg, Kurt-Mothes-Str. 3a, 06120 Halle, Saale, Germany. [2] Biocenter of the University, Martin-Luther-University Halle-Wittenberg, Weinbergweg 22, 06120 Halle, Saale, Germany. [3] Leibniz-Institut für Analytische Wissenschaften -ISAS- e.V., Bunsen-Kirchhoff-Straße 11, 44139 Dortmund, Germany. [4] Segal Cancer Proteomics Centre, Lady Davis Institute, Jewish General Hospital, McGill University, 3755 Côte Ste-Catherine Road, Montreal, QC H3T 1E2, Canada. [5] Biochemistry of Plants, Faculty for Biology and Biotechnology, Ruhr-University Bochum, Universitätsstrasse 150, 44801 Bochum, Germany. ✉email: sacha.baginsky@rub.de

The biogenesis of chloroplasts is essential for plant growth because key metabolic pathways such as photosynthesis, vitamin production, fatty acid and amino acid biosyntheses localize to this organelle. To achieve full functionality, chloroplasts import most of their protein complement from the cytosol. Plants with a defect in the protein translocases at the outer chloroplast (TOC) and the inner chloroplast (TIC) envelope membrane cannot import the proteins necessary for a functional photosynthetic machinery, making protein import essential for chloroplast biogenesis[1]. Deficiencies in protein import also cause transcriptional downregulation of photosynthesis-associated nuclear encodes genes (PhanGs). This is mediated in part by the accumulation of non-imported plastid precursor proteins in the cytosol[2–5]. Precursor proteins are degraded by the ubiquitin proteasome system (UPS), suggesting a regulatory connection exists between retrograde signaling by plastid precursors and their clearance by UPS-mediated precursor degradation[3,5,6].

The UPS is a versatile eukaryotic protein degradation system that utilizes ubiquitination by E3 ubiquitin ligases to control the stability of specific target proteins. Moreover, dynamic changes in the composition of the proteasome complex govern the activity and specificity of proteasome-mediated proteolytic processes[7]. Recognition of poly-ubiquitinated target proteins is mediated by the 26S proteasome, which is built from two main components. One component is the catalytic 20S proteasome (CP) and the other is the 19S regulatory particle (RP), which possibly associate with both ends of the 20S proteasome[7]. The RP consists of a *lid* that is responsible for substrate recognition and binding and a *base* that transfers proteins to the catalytic core. CP and RP may exist and function independently of each other, as exemplified by the upregulation of the CP under conditions of oxidative stress[8,9].

Chloroplast biogenesis and operational control are controlled by the UPS, which targets transcription factors to release the transcriptional block for PhanGs during plant development[10,11]. More recently, direct functions in the turnover of chloroplasts[12] and chloroplast-associated proteins were revealed for components of the chloroplast protein import machinery. For example, a suppressor screen with the plastid protein import mutant 1 (ppi1, deficient in TOC33) identified an E3 ubiquitin ligase, which was termed suppressor of ppi1 locus 1 (Sp1), and was found to ubiquitinate subunits of the TOC complex to target them for degradation[13]. This serves to remodel the import machinery rapidly to accommodate the dynamic requirements of chloroplast proteome adaptations to changing conditions. Similarly, the UPS represses pro-plastid-to-chloroplast differentiation by degrading Toc159 and its import cargo in a DELLA-protein mediated pathway[14].

In the work presented here, we provide a deeper understanding of the processes that integrate the UPS with the regulation of chloroplast protein import and biogenesis. We have recently shown that plastid-precursor proteins accumulate in the cytosol of Toc159-deficient plastids (ppi2)[4,15]. Since precursor proteins are cleared from the cytosol by the UPS[3], we analyzed the effect of proteasome impairment on precursor stability and plastid proteome assembly. We show that double mutants lacking ppi2 and proteasome subunits accumulate more functional photosynthetic complexes by a mechanism that is distinct from the functioning of the E3 ubiquitin ligase Sp1, as it does not operate due to changes in the abundance of TOC components[13]. Instead, we suggest a model whereby mild proteasome impairment affects the turnover of precursor proteins in the cytosol, and this leads to elevated protein import from a larger cytosolic precursor pool. This is particularly effective when protein import is compromised, because decreased precursor turnover results in higher import cargo abundance and permits more time for its

translocation into chloroplasts. Furthermore, these data indicate that under wildtype conditions, synthesis of the photosynthetic apparatus is constrained by proteasomal activity.

## Results

**Proteasomal impairment affects thylakoid membrane stacking.** We selected mutants in regulatory particle non-ATPase subunit 8a (Rpn8a) and its paralogue Rpn8b from the proteasomal *lid* and proteasome subunit alpha-type 1 (PAD1) of the catalytic core and crossed homozygous single mutations of these into the *ppi2* background. In yeast, Rpn8 forms a heterodimer with Rpn11, which is responsible for the removal of polyubiquitin chains prior to substrate degradation in the catalytic core[16]. In Arabidopsis, Rpn8a is the dominant paralogue compared to Rpn8b, while PAD1 and its paralogue PAD2 are integrated in approximately equal amounts into the 20S proteasome complex[17]. Mutants in the E3 ligase Sp1 were tested as a reference, because a defect in this enzyme results in the suppression of the *ppi1* phenotype by direct action on the TOC machinery[13]. The *rpn8a*, *rpn8b*, *pad1* and *sp1* single mutants are phenotypically identical to wildtype supporting functional redundancy among the paralogues (Fig. 1a, Supplementary Fig. 1). The double mutants with *ppi2* are severely

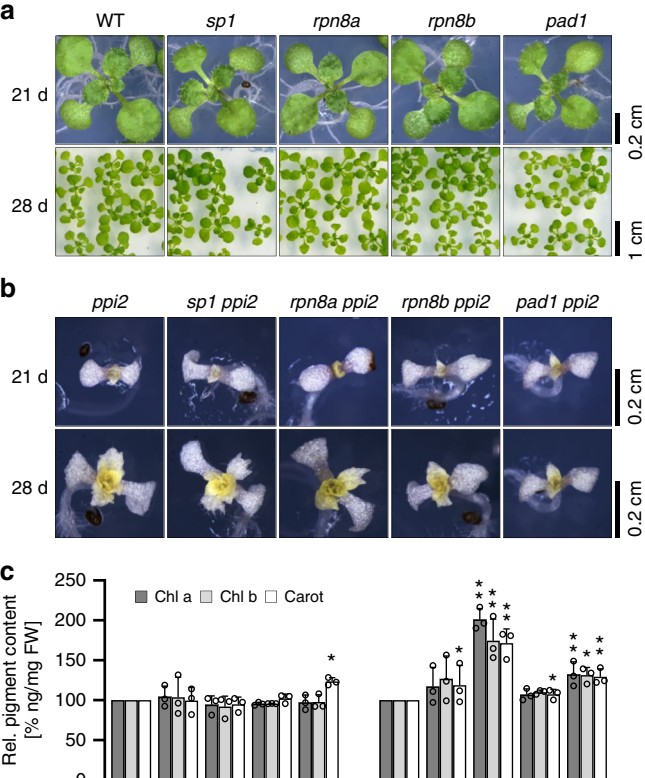

**Fig. 1 Characterization of plant growth for the indicated lines. a** Wildtype (WT), *sp1*, *rpn8a*, *rpn8b*, and *pad1* lines and **b** *ppi2*, *sp1 ppi2*, *rpn8a ppi2*, *rpn8b ppi2*, and *pad1 ppi2* lines were grown for 21 and 28 days under growth light conditions. **c** Pigment content of four-week-old WT, *sp1*, *rpn8a*, *rpn8b*, *pad1*, *ppi2*, *sp1 ppi2*, *rpn8a ppi2*, *rpn8b ppi2*, and *pad1 ppi2* plants were determined as described in the Methods section and statistically evaluated ($n = 3 \times 6$ in case of green plants, $n = 3 \times 30$ in case of albino plants). Significant differences are highlighted by one ($p$-value < 0.05, $T$-test, two-sided) or two stars ($p$-value < 0.005, $T$-test, two-sided) on top of the columns. Error-bars represent SD.

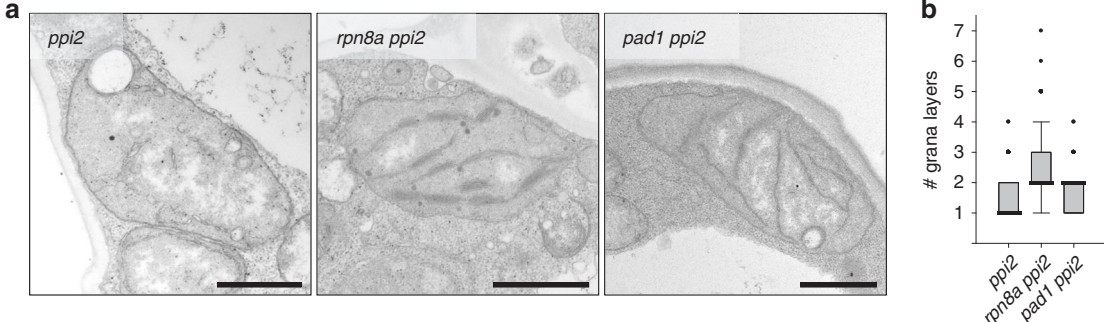

**Fig. 2 Characterization of plastid ultrastructure by transmission electron microscopy (TEM). a** TEM pictures of representative plastids from *ppi2*, *rpn8a ppi2*, and *pad1 ppi2* mutant plants. Scale-bar represents 1 μm. **b** Box-plot representation of the number of stacks counted from three thylakoid stacks per plastid for the three tested genotypes ($n = 153$ plastids, 459 thylakoid stacks). Note the median values for the number of stacks, i.e., *ppi2* = 1, *rpn8a ppi2* = 2, and *pad1 ppi2* = 2 (bold line).

compromised in their growth and chlorophyll content, but their phenotypes are more heterogeneous (Fig. 1b). Photosynthetic pigment measurements show a slight, but significant, increase in carotenoid content in the *pad1* line (Fig. 1c), in *rpn8b ppi2* and in *sp1 ppi2* compared to the reference. While neither *sp1 ppi2*, nor *rpn8b ppi2* show a significant change in chlorophyll levels, the *rpn8a ppi2* and the *pad1 ppi2* double mutants accumulate significantly increased amounts of chlorophyll a, chlorophyll b and carotenoids compared to the *ppi2* single mutant (Fig. 1c). It should be noted that even though, e.g., chlorophyll a content of the *rpn8a ppi2* mutant is nearly doubled compared to *ppi2*, it is still 12-fold lower compared to wildtype.

We considered that elevated pigment levels lead to changes in thylakoid membrane structure and therefore analysed *ppi2*, *rpn8a ppi2,* and *pad1 ppi2* plastids by transmission electron microscopy (TEM) to test this hypothesis. The single mutants were analyzed as a reference and no difference in thylakoid membrane appearance or number of thylakoid stacks was observed compared with the wildtype (Supplementary Fig. 2). In contrast, while *ppi2* showed pro-thylakoid structures that did not stack (median number of stacks is one), we observed thylakoids in *rpn8axppi2* plastids showed significant stacking with a median of two and a maximum of seven layers (Fig. 2a, b, $n = 153$ plastids, 459 thylakoid stacks). The *rpn8a ppi2* mutant furthermore accumulated starch granules, providing indirect but strong support for the functionality of the thylakoid membrane in photosynthesis (see examples in Supplementary Fig. 2). The *pad1 ppi2* mutant had an intermediate phenotype that was characterized by a heterogeneous distribution of thylakoid membrane structures. While some plastids harbored thylakoid membranes with up to four stacks, others essentially resembled the *ppi2* phenotype (Fig. 2a, b, Supplementary Fig. 2). The median number of stacks was *two* demonstrating that more thylakoids are stacked compared to *ppi2*, where the median number of stacks was *one*. Taken together our data show that the *rpn8a* and *pad1* mutations in the *ppi2* background result in a partial suppression of the *ppi2* defects in pigment accumulation and thylakoid stacking Since the plastids in the *rpn8a ppi2* mutant are much more homogenous and the pigment levels are restored to a higher level compared to *pad1 ppi2*, we next characterized the *rpn8a ppi2* double mutant to determine the mechanistic basis for its suppression effects on *ppi2*.

**Improved photosynthetic activity in a proteasomal *lid* mutant.** The above data show that the *rpn8a* mutation partially restores pigment accumulation and thylakoid membrane stacking in the *ppi2* background. To test whether functional photosynthetic

electron transport was restored, we deduced photosynthetic performance from chlorophyll fluorescence parameters. Chlorophyll fluorescence measurements identified higher PSII operating efficiency with lower non-photochemical quenching (NPQ) in the *rpn8a* single mutant, whereas the *rpn8a ppi2* double mutant showed higher PSII operating efficiency, higher maximal PSII quantum yield and higher NPQ over a broad range of photosynthetically active radiation compared to *ppi2* (Fig. 3a, b). Hence, the mutation of *rpn8a* introduced into the *ppi2* background allows the assembly of a stacked thylakoid membrane system resulting in improved photosynthetic electron transport and increased proton translocation into the thylakoid lumen to induce NPQ. The effect of the *rpn8a* mutation on NPQ and PSII operation efficiency in the single mutant was unexpected, as thylakoid stacking and pigment abundance are similar to wildtype (Supplementary Fig. 2). The *rpn8a* single mutant developed more biomass under the chosen growth conditions, which is consistent with improved photosynthetic performance, despite lower resilience against salt stress and a shorter root system (Supplementary Fig. 3). In contrast, elevated photosynthetic activity in the *rpn8a ppi2* mutant did not restore plant growth to an appreciable extent (Fig. 1). The higher NPQ measured for *rpn8a ppi2* compared to *ppi2* suggests, that the pleiotropy of *ppi2* defects precludes efficient utilization of photosynthetic activity thus resulting in the scavenging of extra excitation energy in *rpn8a ppi2*, and not in a biomass increase (Fig. 3a).

We next examined whether increased pigment levels and thylakoid stacking in the *rpn8a ppi2* double mutants resulted in increased accumulation of thylakoid membrane proteins. Immunoblotting for selected thylakoid membrane proteins and outer envelope translocon (TOC) subunits identified higher amounts of light-harvesting complex protein 4 (Lhcb4.1) in the *rpn8a ppi2* double mutant, while no effect on the abundance of Toc75 and Toc132 was detectable (Fig. 4a, b). In contrast, the *sp1 ppi2* double mutant did not accumulate higher levels of Lhcb4, despite having increased levels of Toc75 and Toc132 (Fig. 4a, b). The *rpn8a* and *sp1* single mutants showed no change in the accumulation of photosynthetic proteins compared to wildtype, despite increased accumulation of Toc75 in *sp1* (Fig. 4a, b).

Protein quantification on a proteome-wide scale[18] in *sp1* and *rpn8a* single mutants supported the lack of significant quantitative differences in the amount of thylakoid membrane protein complexes compared to wildtype (Fig. 5a, Supplementary Data 1). Rather, there was a slight, but significant, increase (1.96-fold compared to wildtype, p-value 0.015, *T*-test, two-sided) in the amount of the small subunit of ribulose 1,5-bisphosphate carboxylase (RbcS) in the *rpn8a* mutant. The quantitative proteome data obtained with the double mutants confirmed that

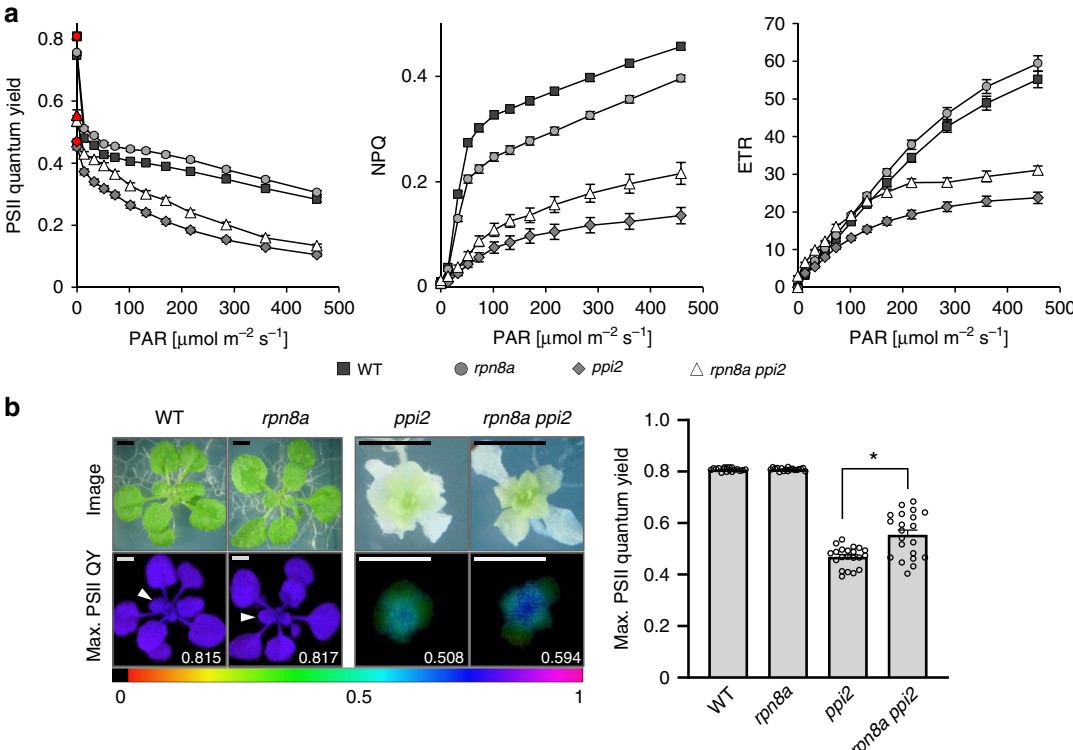

**Fig. 3 Photosynthetic performance determined by chlorophyll fluorescence measurements. a** Imaging PAM determination of PSII quantum yield and activity (left graph), non-photochemical quenching (NPQ) (middle graph) and electron transport rate (ETR) (right graph) for wildtype (WT), *rpn8a*, *ppi2*, and *rpn8a ppi2* ($n = 18$ in case of green plants, $n = 21$ in case of albino plants), error-bars represent SEM. **b** Graphical representation of maximum PSII quantum yield (left: fluorescence images, right: column representation of determined max. PSII quantum yield ($n = 21$, *T*-test, two-sided, *p*-value < 0.001 (one star)) Error-bars represent SEM, the scale bars represent 0.2 cm.

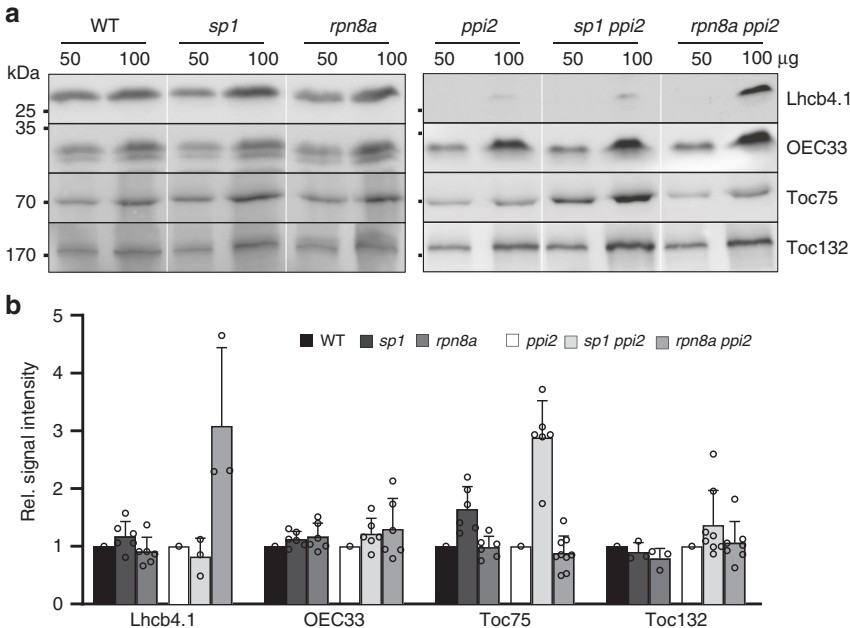

**Fig. 4 Determination of protein abundance in the indicated plant lines. a** Immunoblotting with antibodies against Lhcb4, OEC33, Toc75, and Toc132 with 50 and 100 µg plant extracts. **b** Quantitative determination of the indicated proteins from western blot analyses. Data are derived from at least three biological replicates. Genotypes tested were wildtype (WT), *sp1, rpn8a, ppi2* single mutants and *sp1 ppi2* and *rpn8a ppi2* double mutants. The membrane was decorated with antibodies against the indicated proteins and the intensity was determined by Image J after normalization against the amido-black stain of the membrane. Error-bars represent SD. Note that the number of data points provided for the Lhcb4 antibody is three as opposed to six for the other antibodies. This is because no Lhcb4 signal was detected in several replicates with *ppi2*, so that a relative signal intensity could not be provided.

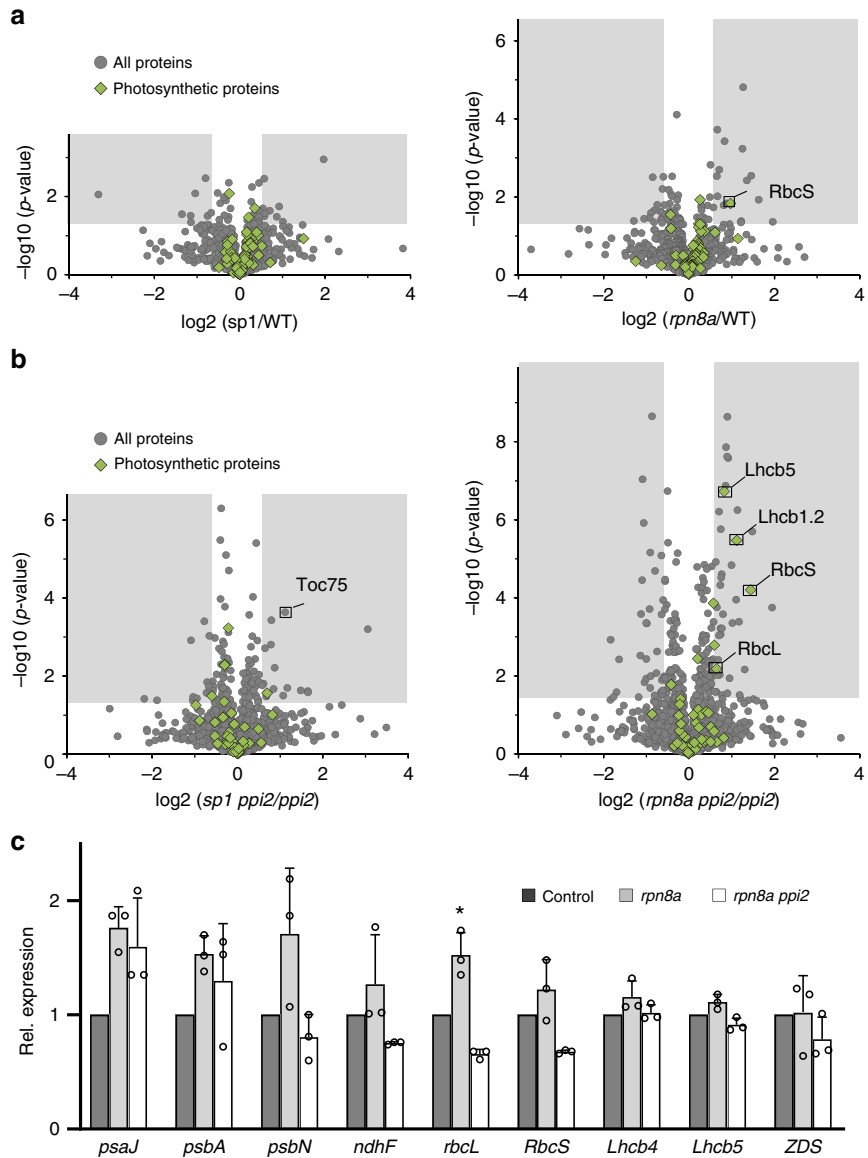

**Fig. 5 Quantitative proteomics and real-time PCR analyses. a, b** Volcano plot of the results from quantitative proteome profiling with the different genotypic lines. Data on the x-axis represent the Log2-ratio between the absolute abundances of proteins identified from the *sp1* (left plot) and *rpn8a* (right plot) single mutants against wildtype (**a**) and the Log2-ratio of absolute abundances between the *sp1 ppi2* (left plot) and the *rpn8a ppi2* double mutants against *ppi2* (**b**). The y-axis represents the Log10 of the p-value from a two-sided T-test. Significant changes are indicated by the gray box (ratio > 1.5 times different, p-value < 0.05, T-test, two-sided). All data are based on three biological replicates that were measured in three technical replicates each. **c** Quantitative real-time PCR results obtained with the transcripts of selected genes for photosynthetic proteins. Significant differences from three biological replicates are highlighted by a star (T-test, two-sided, p-value < 0.05), error-bars represent SD.

the *rpn8a* mutation increased photosynthetic protein accumulation without any quantifiable effect on the TOC machinery. While the *sp1 ppi2* double mutant showed significantly increased concentrations of Toc75 (2.18-fold compared to *ppi2*, p-value $2.4 \times 10^{-4}$, T-test, two-sided), Toc34 and Toc33 (these were exclusively identified in *sp1 ppi2*), there was no effect on the accumulation of photosynthetic proteins (Fig. 5b, Supplementary Data 2). Thus, the abundance of the TOC components (other than Toc159) did not limit protein import in the *ppi2* mutant. The *rpn8a ppi2* mutant accumulated significantly higher amounts of light harvesting complex II (LHCII), photosystem II (PSII) and the ribulose-1,5-bisphosphate carboxylase (RubisCO) complex without showing any change in the abundance of the TOC machinery (Fig. 5b, Supplementary Data 2). Several subunits of photosystem I, PSII, light harvesting complex I, LHCII and PetA from the cytochrome

b6-f complex were exclusively identified in *rpn8a ppi2*, and are therefore not represented in the Volcano plot (Supplementary Data 2, Fig. 5b). Their exclusive detection in the double mutant is consistent with their elevated abundance in the *rpn8a* mutant background. Thus, the TOC components are sufficient to accommodate the increased accumulation of photosynthetic proteins in the *rpn8a ppi2* double mutants.

Importantly, the effect of the *rpn8a* mutation on the accumulation of nuclear-encoded photosynthetic proteins in the *ppi2* background originates at the post-transcriptional level, as indicated by the lack of transcriptional induction of key photosynthetic genes (Fig. 5c). This is therefore distinct from the reported action of the proteasome on nuclear transcription factors regulating plastid biogenesis[10,11,19] and from the retrograde signal transduction chains operating via precursor-protein

accumulation[5]. Intriguingly, the expression of plastid-encoded genes for photosynthetic proteins such as RbcL or PsbA was not different between *ppi2* and *rpn8a ppi2*, while their protein products accumulated to significantly higher levels in the double mutant (Fig. 5b, c, Supplementary Data 2). Thus, we observed a concerted accumulation of chloroplast- and nuclear-encoded subunits e.g. for the RubisCO complex. This phenomenon is known as CES (Controlled by Epistasy of Synthesis) and is accomplished either by the rapid proteolytic degradation of unassembled subunits, or by adjusting the rate of synthesis of plastid-encoded subunits in response to the availability of their assembly partners[20]. In the *rpn8a* single mutant, increased abundance of RbcS (Fig. 5a, Supplementary Data 1) was matched by increased expression of plastid-encoded RbcL, suggesting that here, CES may also operate at the transcriptional level (Fig. 5c).

**Changes in proteasome composition in *rpn8a* mutants.** To characterize the mechanistic basis for the suppression phenotype caused by the *rpn8a* mutation, we analyzed the composition of the proteasome in the *rpn8a* single mutant and in the *rpn8a ppi2* double mutant; as a control we tested the *sp1* and *sp1 ppi2* mutants. In both *rpn8a* single and double mutant lines, the 20S catalytic core had a significantly higher abundance compared to the reference, while the total abundance of the *base*-component of the proteasome was unchanged. In contrast, the abundance of the *lid*-subunit in the double mutant was significantly decreased to around 70% of the level in the *ppi2* mutant. No *lid* subunit was identified in the *rpn8a* single mutant (Fig. 6a). We identified Rpn8b, the homolog of Rpn8a, only in the *rpn8a ppi2* double mutant, where it accumulated to around 70% of Rpn8a levels in the *ppi2* mutant, and this correlated with the overall amount of the *lid* (Supplementary Data 2). This suggests that the amount of Rpn8 subunits limits the assembly of the *lid* and that Rpn8b can functionally replace Rpn8a, albeit full functional complementation is limited by the lower amount of Rpn8b compared to Rpn8a. As anticipated, the *sp1* mutation had only a minor effect on proteasome composition, which was restricted to decreased abundance of the *lid* in the single mutants and a slightly elevated abundance of the 20S proteasome in the *sp1 ppi2* double mutant (Fig. 6a). Despite being significant, both proteasome modifications are by far less prominent than those in the *rpn8a* mutants.

To test the extent to which the above changes in proteasome composition in the *rpn8a* mutants affects proteasome activity, we measured the relative chymotrypsin-like proteolytic activity of the proteasome in total plant extracts using a fluorogenic substrate[21]. This assay measures the activity of both the 20S and the 26S proteasome. Absolute chymotrypsin-like activity was identical between the *rpn8a ppi2* double mutant and the *ppi2* single mutant, while changes in proteasome composition and activity were observed as a slightly elevated resistance to MG132 (Fig. 6b). This was supported by a quantitative assay[22] that indicated increased tolerance of the mutant toward the inhibitor, thus supporting elevated 20S proteasome activity in the *rpn8a* mutants (Fig. 6c). Together, these data show that the compensatory upregulation of the 20S proteasome, which was observed in the proteomics data, reconstitutes overall proteasome activity. In the presence of MG132, the access of the mutants to a larger pool of 20S proteasome explains the increased tolerance toward the inhibitor.

Overall proteasome activity was indistinguishable between the *rpn8a* mutants and the reference. Provided that the 26S proteasomal *lid* subunits were significantly decreased, while the 20S proteasome was significantly increased in the *rpn8a* mutants (Fig. 6a), this suggested that there was a shift in the relative activity of 20S and 26S proteasome in vivo. This proteasomal

reorganization is equivalent to a shift between different pathways for proteasomal function[23]. To test the extent to which this shift caused proteasomal stress, we examined the expression of typical markers of the proteasomal stress regulon[24] in *rpn8a* single and *rpn8a ppi2* double mutants. Here, expression of the genes encoding proteins for proteasome activity and assembly, such as CDC48a and NAS6, the *lid* subunits Rpn8b and Rpn5a, the *base* subunits Rpn10 and Rpt2a, and the 20S subunit PAA2 (Fig. 6d) were upregulated at the transcriptional level. Thus, mutant plants expressed typical stress markers to counteract the decrease in 26S proteasome activity. The discrepancy between transcriptional upregulation and quantitative protein accumulation for the components of the *lid* and *base* (Fig. 6a and d) shows that *rpn8a* mutants are impaired in 26S proteasome assembly (Fig. 6e).

To test for the effect of inhibitor-induced proteasomal stress on photosynthetic protein abundance, we treated wildtype and *ppi2* plants with MG132 and compared the proteomes of inhibitor-treated plants with the proteomes of *rpn8a* single and *rpn8a ppi2* double mutants. Both, inhibitor-treated plants and *rpn8a* mutants accumulated significantly elevated levels of the 20S proteasome compared to their reference plants (Fig. 6a). In contrast, the *lid* and *base* subunits accumulated to higher levels in MG132-treated plants, while the level of the *lid* subunit remained unchanged and the *base* subunit had significantly lower levels in the *rpn8a* single mutant and in the *rpn8a ppi2* double mutant. Importantly, MG132-treated plants showed no increase, but rather a decrease, in the abundance of photosynthetic proteins (Supplementary Data 3 and 4). Thus, a general reduction of proteasomal activity has an effect on chloroplast protein abundance that is different from the *rpn8a* mutation. This is not surprising because the 20S proteasome removes damaged proteins under conditions of oxidative stress[9] and inhibition of its activity by MG132 interferes with this important function[25].

Consistently, *rpn8a* mutants with elevated levels of active 20S proteasome expressed lower amounts of typical oxidative stress-marker proteins compared to their reference (Supplementary Fig. 4). Intriguingly, low-light conditions mimicking decreased oxidative stress resulted in elevated levels of photosynthetic proteins in both, *ppi2* and in *rpn8a ppi2* mutants (Supplementary Fig. 4) suggesting that decreased oxidative stress due to elevated 20S levels most likely contributes to the suppression phenotype. However, differences in photosynthetic protein and pigment accumulation between *ppi2* and *rpn8a ppi2* remained significant under low-light conditions (Supplementary Figs. 4 and 5). Thus, elevated 20S levels are not solely responsible for the suppression phenotype in the *ppi2* background, rather, a moderate decrease in 26S proteasome activity appears to be required.

**Elevated precursor abundance in the cytosol of *rpn8a ppi2*.** The 26S proteasome is involved in the degradation of non-imported plastid precursor proteins, as has been demonstrated for the Lhcb4, LTA2 and the PDH E1 α precursor[3,6]. MG132 stabilized the RbcS-, the ferredoxin (Fd)-, and the RNP29- precursors fused to GFP in protoplast assays (Fig. 7a, Supplementary Fig. 6), demonstrating that proteasomal degradation of precursor proteins in the cytosol applies to a broader set of target proteins. Intriguingly, the absolute abundance of mature ferredoxin and RbcS accumulating in import-impaired *ppi2* plastids was higher when precursors were stabilized with MG132 in protoplast assays, providing an explanation for the effect of proteasome impairment on thylakoid membrane assembly (Fig. 7a, Supplementary Fig. 6). To test for the amount of plastid precursor proteins in vivo, we analyzed precursor abundance in the *rpn8a ppi2* double mutant and in the *ppi2* single mutant by N-terminal proteome profiling using ChaFRADIC in combination with iTRAQ quantification[26].

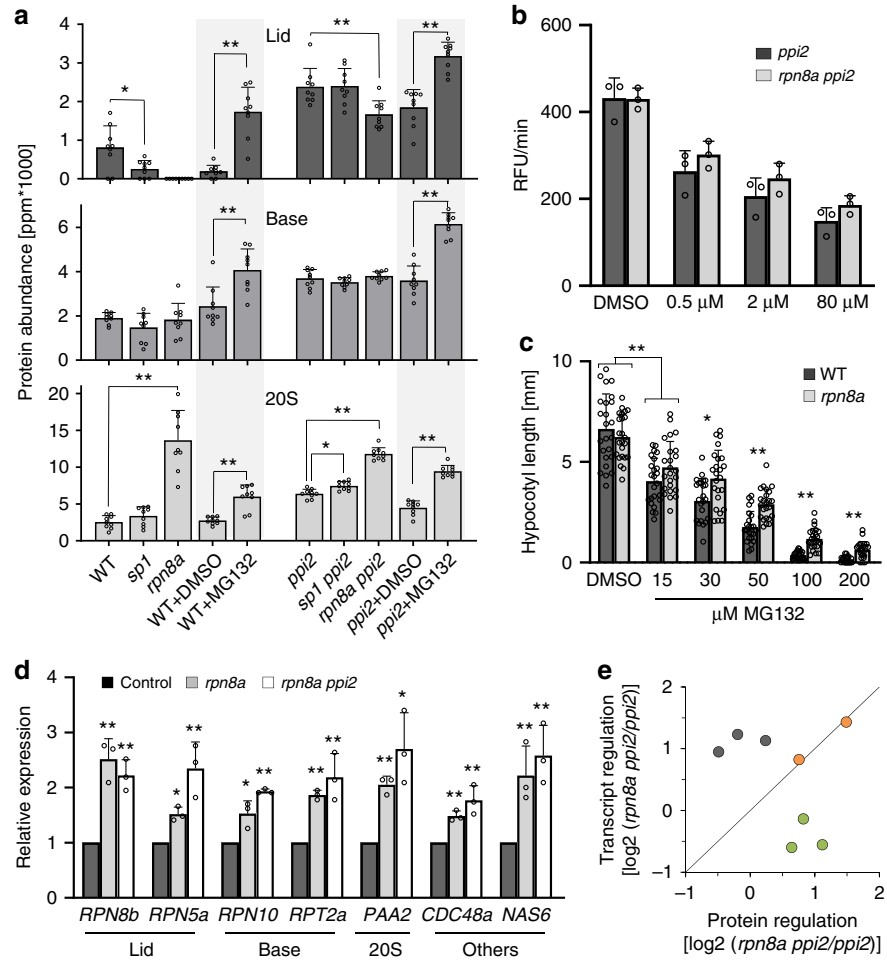

**Fig. 6 Characterization of proteasome accumulation and activity. a** Quantitative determination of proteasome subunits by proteome-wide protein quantification[18] of the indicated genotypes and after the indicated treatment (DMSO control, +MG132). The ppm values for all subunits belonging to the *lid* (upper panel), the *base* (middle panel) and the 20S catalytic core (lower panel) from three biological and three technical replicates were summed up and the results are presented along with their standard deviation (SD). The annotation of subunits to the *lid*, the *base* and the core is as published previously[17]. Significant differences are indicated with one (*p*-value < 0.05, *T*-test, two-sided) or two (*p*-value < 0.01) stars (two-sided *T*-test). **b** Chymotrypsin activity of the proteasome reported in RFU/min as determined from extracts of *ppi2* and *rpn8a ppi2* mutants (error-bars represent SD, note that the differences are not significant). **c** Hypocotyl length determination from wildtype and *rpn8a* single mutants from 5-day old etiolated seedlings, presented as the mean from 24 plants, grown in the presence of the indicated amount of MG132 (error-bars represent SD). Significant differences are indicated with one (*p*-value < 0.005) or two (*p*-value < 0.001) stars (two-sided *T*-test). **d** Quantitative real-time transcript abundance determination for different *lid, base*, 20S subunits and regulatory components ("other") from wildtype and *rpn8a* mutant plants. Error-bars represent SD. Significant differences are indicated with one (*p*-value < 0.05) or two (*p*-value < 0.01) stars (two-sided *T*-test). **e** Relationship between transcriptional regulation (*y*-axis: Log2-transcript ratio from the double mutant in relation to the *ppi2* single mutant) and protein abundance (*x*-axis: Log2-protein abundance ratio from the double mutant in relation to the *ppi2* single mutant) for three proteasome subunits (gray), two regulatory factors (orange) and three photosynthetic proteins (green) for comparison.

Eighteen plastid precursor proteins with a clearly defined transit peptide were identified from *ppi2* single and *rpn8a ppi2* double mutants. Three of these have a significantly higher abundance in the double mutant (ZDS, NAD(P)-binding Rossmann-fold superfamily protein, aspartate aminotransferase) and all precursor proteins functionally connected with photosynthesis (ZDS, PsbO-1, PsbO-2, RbcS, Lhcb6, PGK1, PGK-family, PIFI) were on average 1.2 times more abundant in the double mutant, with the precursor of zeta carotene desaturase (ZDS) being significantly increased by a factor of 1.46 and a p-value of 0.02 (*T*-test, two-sided) (Fig. 7b, Supplementary Data 5). This difference in precursor abundance is controlled at the post-transcriptional level, as, e.g., ZDS transcript levels were unchanged (Fig. 5c).

The abundance of cytosolic HSPs was moderately increased in the *rpn8a ppi2* double mutant, which is consistent with elevated precursor levels in the cytosol (Fig. 7c). The identified HSPs

belong to the HSP70-family, i.e. Hsc70 (At5g02500), which is upregulated by a factor of 1.27 (*p*-value 0.0009, *T*-test, two-sided), HSP70 (At1g79920) by a factor of 1.2 (*p*-value 0.008, *T*-test, two-sided) and HSP70b by a factor of 2.6 (*p*-value 0.04, *T*-test, two-sided) (Fig. 7c). The levels of HSP90-family proteins, which play a role in propagating the retrograde signal emanating from plastid precursor proteins[5], were not significantly different between the *ppi2* and *rpn8a ppi2* mutant, but we observed a significant induction of cytHSP90 (At5g56000) in the *rpn8a* single mutant (factor 3.9, *p*-value 0.04, *T*-test, two-sided, Supplementary Data 1). This induction was considerably lower than the one reported for a *gun1clpc1* double mutant in which retrograde signaling was induced[5]. Together, a higher precursor level in the cytosol of *rpn8a* mutants, resulting in increased import by an impaired plastid protein import system, is a plausible explanation for the observed phenotypes.

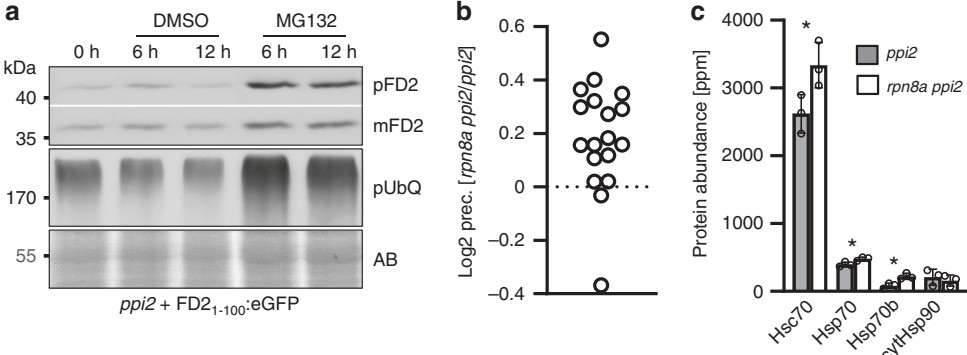

**Fig. 7 Functional basis for the observed phenotype. a** Protoplast import assay and subsequent extraction of proteins for immunoblotting. Protoplasts were transformed with the precursor of ferredoxin 2 (1–100 amino acids) fused to GFP and treated with DMSO and/or MG132 before protein extracts were obtained. Blots were either decorated with the GFP-antibody identifying the precursor of ferredoxin 2 (pFd2) and the mature form of ferredoxin 2 (mFd2), or with a pUbQ antibody identifying poly ubiquitinated proteins (pUbQ). The loading control is presented as amido black stain of the membrane (AB). **b** ChaFRADIC identification of N-terminal peptides identified 18 precursor proteins. The results of relative precursor quantification by iTRAQ are presented as the Log2-ratio between precursor abundance in the *rpn8a ppi2* double mutant compared to *ppi2* as the mean value from three biological replicates. For more information about the precursors see Supplementary Data 5. **c** Abundance of cytosolic heat shock proteins in *ppi2* and *rpn8a ppi2* mutants determined from three biological replicates. Significant differences are highlighted by a star (*T*-test, two-sided, *p*-value < 0.05).

## Discussion

In this study we show an influence of proteasome composition on photosynthetic performance of protein import-impaired plastids and wildtype chloroplasts. Mutations in the *lid* subunit Rpn8a and the core subunit PAD1 partially restore pigment content and photosynthetic capacity of *ppi2* mutants, while a mutation in the E3 ligase Sp1, previously identified as a suppressor of *ppi1*, does not affect the *ppi2* phenotype to the same extent (13, Supplementary Data 2). In the absence of Toc159, the molecular phenotype of *sp1* is characterized by increased abundance of other subunits of the TOC complex such as Toc33, Toc75 and Toc132, which does not influence the accumulation of photosynthetic or other chloroplast proteins (Figs. 4 and 5, Supplementary Data 1 and 2). It is therefore intriguing that the *rpn8a ppi2* double mutant accumulates significantly increased levels of functional photosynthetic protein complexes, as shown by chlorophyll fluorescence measurements, PSII efficiency, NPQ, thylakoid stacking and starch accumulation (Figs. 2–5). This occurs without any change in the abundance of TOC components (Supplementary Data 1 and 2), demonstrating that Sp1 and proteasome mutants affect the phenotype of protein import mutants by distinct mechanisms.

Assuming that cytosolic precursor proteins elicit retrograde signaling[5], it is possible that elevated levels of the 20S proteasome in the *rpn8a* mutants could result in a more efficient clearance of precursors, which eliminates the transcriptional downregulation of PhanGs (Fig. 6a–c). This is plausible because the transit-peptide creates a disordered region that makes precursors an efficient natural target for 20S proteasomal degradation[9]. However, our data show that this is not the mechanism responsible for the phenotype observed here. First, there is no effect of the mutation on the expression of key photosynthetic proteins at the transcriptional level (Fig. 5c). Secondly, the depletion of PAD1, an α-subunit of the 20S proteasome, has a similar effect on pigment levels in the *ppi2* background as *rpn8a*. Thirdly, the level of precursors is not reduced in the *rpn8a* mutants but rather is increased (Fig. 7b). Finally, cytosolic HSPs are significantly more abundant in the *rpn8a ppi2* double mutant, which is consistent with elevated cytosolic precursor levels (Fig. 7c, Supplementary Data 1, 2 and 5).

Recent analyses revealed the regulation of chloroplast protease abundance by the cytosolic E3 ligase AtCHIP, which could indirectly affect the assembly of thylakoid membrane protein complexes[27]. AtCHIP is a component of the UPS and its overexpression results in decreased accumulation of Ftsh1 and Ftsh2, two proteases that are integral to the chloroplast protease network. However, this is specific for high-light conditions and no effect of AtCHIP overexpression on chloroplast proteases was observed under standard growth-light conditions[27]. Consistent with our suggestion that this effect is specific for stress conditions, we do not find any significant general effect of the *rpn8a* mutation on the chloroplast protease network, neither in the wildtype, nor in the *ppi2* background (Supplementary Data 1 and 2). One exception, however, is FtsH5, which accumulates to significantly higher levels in both *sp1 ppi2* and *rpn8a ppi2* double mutants. FtsH5 localizes to the thylakoid membrane[28] and its increased abundance is therefore expected to result in increased thylakoid membrane protein turnover. Thus, its elevated abundance is unrelated to the phenotype observed here.

The dynamic regulation of organellar protein abundance by the cytosolic UPS was also observed for mitochondria in yeast. Here, a significant fraction of precursor proteins destined for the inter membrane space (IMS) is constantly degraded by the proteasome. Proteasome inhibition leads to increased protein import and elevated accumulation of imported proteins inside mitochondria, suggesting that the proteasome continuously degrades functional precursors[29–31]. Thus, when assembly of the 26S proteasome as observed here (Fig. 6e) is impaired, the decreased rate of precursor turnover provides an explanation for the accumulation of mature photosynthetic proteins in protein import-impaired plastids, which happens without transcriptional induction (Fig. 5c). Despite the effect of the mutation on precursor abundance being subtle (Fig. 7b), it is conceivable that permanently increased precursor half-lives during plant development and growth may result in a cumulative effect on photosynthetic protein accumulation, especially when protein import is compromised. This explanation is supported by the fact that the influence of mild proteasome impairment on import-deficient plastids is not restricted to *ppi2* mutations, but is also visible with other protein import mutations, such as *ppi1* and *toc75-III*[13]. Thus, the degradation of functional precursor proteins in the cytosol is a common process that affects cell organelles with impaired protein import capacity.

Together, our data reveal a general effect of mild proteasome impairment on the chloroplast proteome that functions by increased precursor stability. Photosynthetic proteins are those

most prominently affected, but identifying potential effects on other precursor types requires further study. In analogy to yeast mitochondria, chloroplast precursor degradation occurs continuously, even in the absence of an import defect, as shown by the beneficial effect of the *rpn8a* mutation on NPQ and RbcS levels in the wildtype background (Figs. 3a, 5a). This suggests that photosynthesis is permanently constrained by proteasomal activity, possibly to counteract potentially deleterious effects of electron transport. It is thus conceivable that stabilization of precursor proteins by other means, e.g., by modification of presently unrecognized N-terminal degrons, may lead to increased plastid protein accumulation. This could be of interest for biotechnological applications. Protein import, cytosolic translation and proteasome activity form a regulatory triad in yeast that is tightly connected with mitochondrial quality control and protein homeostasis[29–31]. Our analysis, and the mutants generated within its course, paves the way for further characterization of this regulatory system in plant cells and will allow us to define how it affects chloroplast biogenesis and function.

## Methods

**Plant material, accession numbers, and growth conditions**. We obtained the following mutant lines from the NASC collection, i.e., SALK_151595C (*rpn8a*), SALK_128568C (*rpn8b*), SALK_047984C (*pad1*) in the Col-background (wildtype reference). The *sp1* (SALK_063571) and *ppi2* mutants (Toc159, CS11072 introgressed into the Columbia-0 ecotype) are as previously reported[9,32]. Double mutants *rpn8a ppi2*, *sp1 ppi2*, *rpn8b ppi2*, and *pad1 ppi2* were crossed with the above referenced genotypes using heterozygous *ppi2* plants. Primers used for the mutant characterization are provided in Supplementary Data 6. After 2 days of stratification at 4 °C the plants were grown on half-strength Murashige and Skoog (M&S) medium supplemented with 0.8% (w/v) plant agar (Duchefa) and 0.8% (w/v) sucrose under short day conditions. For inhibitor treatments, plants were transferred after 7d into liquid MS medium with 0.8% (w/v) sucrose and cultivated for the indicated time. For some experiments with wildtype and single mutants, plants were cultivated on soil for the indicated time. The plant proteins investigated throughout this work have the following accession numbers: Rpn8a - AT5G05780; Rpn8b - AT3G11270; PAD1 – AT3G51260; Rpn5a – AT5G09900; Rpn10 – AT4G38630; Rpt2a – AT4G29040; PAA2 – AT2G05840; CDC48a - AT3G09840; NAS6 - At2g03430; Sp1 - AT1G63900; Toc159 - AT4G02510.

**Growth assays**. Root length was determined after 7 and 17 days from plants grown under short-day conditions on plate. For the determination of fresh weight and pigment concentration seeds were grown for 28 days in liquid culture under short-day conditions. Plants were harvested and the fresh weight of individual plants was determined with a SartoriusTM BP211D micro scale. Pigment concentration of green or pale-green plants was determined for individual plants, while five plants were pooled for every measurement with albino plants. The amount of chlorophyll a, chlorophyll b and carotenoids was determined as described by Lichtenthaler and Buschmann[33].

**Transmission electron microscopy (TEM)**. Leaf segments were fixed with 3% glutaraldehyde (Sigma) in 0.1 M sodium cacodylate buffer pH 7.2 (SCB) for 4 h, washed in SCB, postfixed for 1 h with 1% osmiumtetroxide (Carl Roth) in SCB, dehydrated in a graded series of ethanol and embedded in an epoxy resin. Ultrathin sections (80 nm) were observed with a LIBRA 120 (Carl Zeiss Microscopy, Oberkochen, Germany) transmission electron microscope (acceleration voltage 120 kV). Electron micrographs were taken with a dual-speed on axis SSCCD camera (BM-2k-120; TRS, Moorenweis, Germany) using the iTEM software from Olympus SIS (Münster, Germany).

**Imaging PAM**. Seedlings were grown on 0.5 x MS-Medium supplemented with 3% (w/v) sucrose for 28 days under short-day conditions. Leaf seven or eight from normal-light and leaf two or three from low-light grown seedlings (in either case the second youngest leaf) were selected for the measurements. The albino plants (*ppi2* and *rpn8a ppi2*) were grown for 48d under the same conditions. Fluorescence was measured with an imaging-PAM chlorophyll fluorometer (Walz). After determination of max. PSII quantum yield obtained with a saturating light-pulse on 20 min dark-adapted seedlings, we measured PSII quantum yield after 20-s adaptation of the seedlings to increasing light intensity. All parameters were calculated using ImagingWin (Walz).

**Plasmid construction**. The vector backbone of all plasmids was pRT100 Ω/Not/ Asc[34] containing the coding sequence of eGFP (Clontech). Wild-type Arabidopsis cDNA was used as template to amplify the coding sequence for the first 100 amino

acids of the proteins of interest by PCR (Supplementary Fig. 1 and Supplementary Data 6). Sequence was first ligated into pCR2.1®-TOPO® vector (TA cloning®, Invitrogen) and subsequently cloned into the target vector in-frame upstream of the eGFP sequence. The plasmids pRT100 Ω/Not/Asc_eGFP as well as pRT100 Ω/ Not/Asc_FNR1 − 55:eGFP were provided by R. B. Klösgen.

**RNA extraction and real-time PCR**. RNA was extracted from ~100 mg plant material with the NucleoSpin® RNA Plant isolation kit (Macherey & Nagel). For the synthesis of cDNA, 4 µg total RNA were reverse transcribed with a cDNA synthesis kit (Thermo Fisher Scientific) and oligo-(dT)18 oligonucleotides for cloning and random hexamer oligonucleotides for quantitative transcript analyses. Primers for real-time PCR were optimized with the help of Primer Blast. For an assay of 20 µl we used 1-µl cDNA (1:5) with 10 µl IQTM SYBR Green Supermix (Bio-Rad Laboratories) and primer oligonucleotides at a final concentration of 200 nM. The reaction was carried out in an iCycler (Bio-Rad Laboratories) in three biological replicates that were measured in three technical replicates each. Transcript data were normalized to actin 7 (At5g09810) and the quantities of transcripts were calculated as 2(-Delta Delta C(t)) values[35].

**Protease activity assays**. Proteasome activity was measured from frozen seedlings that were ground in 200 µl extraction buffer (2 mM ATP, 50 mM HEPES-KOH (adjusted to pH 7.2 with 1 N KOH), 2 mM DTT, 0.25 M sucrose), centrifuged at 17,000 × $g$ for 10 min at 4 °C and the protein concentration adjusted to a final protein concentration of 1 mg/ml. Twenty-five micrograms of plant extract were transferred into the wells of a black walled 96-well plate and subsequently mixed with 220 µl assay buffer (100 mM HEPES-KOH (adjusted to pH 7.8 with 1 N KOH), 5 mM MgCl2, 10 mM KCl, 2 mM ATP). After the addition of Suc-LLVY-AMC (dissolved in DMSO) as substrate to a final concentration of 100 µM, we measured the release of amino-methyl-coumarin at 30 °C (A360ex/A460em) on a fluorescent plate reader every 1.5 min over a period of 120 min. Activity was determined as relative fluorescence units (RFU) per minute, after subtracting the values from a blank control. All assays were performed in three biological replicates and measured on a Tecan Spark microplate reader[21].

**Protoplast assays**. Plants were harvested immediately after the end of the dark period, with root tissue excluded. The plant material was cut into shreds and transferred to enzyme solution [400 mM sorbitol, 5 mM MES (pH 5.6), 8 mM CaCl2, 1.5% (w/v) Cellulase Onozuka R-10 (Serva), 0.375% (w/v) Macerozyme R-10 (Serva)], vacuum infiltrated and incubated for 4 h at room temperature in the dark. Protoplasts were released by gentle shaking. After filtration (100 µm BD Falcon™ cell strainer) the number of protoplasts was estimated with a Neubauer chamber. The protoplasts were settled by centrifugation (100×g, 5 min) and adjusted to 2 × 10^6 protoplasts per ml in 230 mM NaCl, 187 mM CaCl2, 7.5 mM KCl, 7.5 mM glucose, 2.3 mM MES (pH 5.6). After chilling on ice for 30 min the protoplasts were settled again by centrifugation (100 × $g$, 5 min) and transferred into 0.4 M sorbitol, 15 mM MgCl2, 5 mM MES (pH 5.6) maintaining the amount of 2 × 10^6 protoplasts per ml. One hundred microliter of protoplast solution was mixed up with 10 µg plasmid DNA each and 110 µl PEG solution [60% (w/v) PEG4000 (Fluka), 0.3 M sorbitol, 0.15 M Ca(NO3)2] and incubated for 20 min at room temperature. Protoplasts were washed twice with 230 mM NaCl, 187 mM CaCl2, 7.5 mM KCl, 7.5 mM glucose, 2.3 mM MES (pH 5.6) and once with protoplast culture medium (M&S medium, 350 mM sorbitol, 50 mM glucose, 3 mM CaCl2, pH 5.8) including 0.1 mg/ml Ampicillin. Transformed protoplasts were stored in protoplast culture medium in darkness.

**SDS PAGE and Western-blotting**. Protoplast proteins were extracted by adding SDS sample buffer [50 mM Tris/HCl (pH 6.8), 2% (w/v) SDS, 10% (v/v) Glycerol, 0.1 M DTT, 0.04‰ Bromphenol blue] and heating the extract for 5 min at 90 °C. Plant proteins were extracted from shock-frozen seedlings that were grinded and resolved in Rensink buffer [100 mM NaCl, 50 mM Tris/HCl (pH 7.5), 0.5% (v/v) Triton X-100, 2 mM DTT] including plant protease inhibitor cocktail (Sigma-Aldrich). Fifty or hundred microgram protein extract as indicated was separated by SDS-PAGE on 12% polyacrylamide gels and transferred by semi dry blotting using PerfectBlueTM (VWR International GmbH) onto polyvinylidene difluoride (PVDF) membranes. Immunodetection of proteins was done using enhanced chemiluminescence, and images were obtained by the Fusion Fx7 image-acquisition system (Peqlab). The following antibodies were used in the following dilutions: antiGFP (MBL 598) 1/5000, antiLhcb4 (Agrisera (AS04 045)) 1/7000, antiActin (Sigma-Aldrich) 1/5000, antiToc75 and antiToc132 (generously provided by Prof. D.J. Schnell, Michigan State University) both 1/2000, antiUbQ11 (Agrisera (AS08 307) 1/10,000 and antiOEC33 (generously provided by Prof. R. B. Klösgen, MLU Halle-Wittenberg) 1/5000.

**Protein identification by mass spectrometry**. The quantitative proteome was determined from wildtpye (Col-0), *rpn8a*, *ppi2* and *rpn8a ppi2* mutants from 100 mg plant material. To assess the influence of MG132 on the plant proteome, we grew plants in liquid culture for 28d under short-day conditions. MG132 was added to a final concentration of 50 µM in DMSO and incubated for 30 h. Again, 100 mg plant material from wildtype (DMSO-control), *ppi2* (DMSO-control),

wildtype + MG132 and ppi2 + MG132 were used for protein extraction. Plant material was homogenized in Rensink extraction buffer (50 mM Tris/HCl pH 7.5, 100 mM NaCl, 0.5% (v/v) TritonX-100, 2 mM DTT and protease inhibitor cocktail (Sigma-Aldrich)). The extract was cleared by centrifugation and subsequent chloroform/methanol precipitation. LC separation and HD-MSE data acquisition was performed as described by Helm and colleagues[18]. Precisely, for every probe we digested 100 μg protein in solution with RapiGestTM with 1 μg Trypsin (Promega) over night. Peptide pellets were dissolved in 2% (v/v) ACN, 0.1% (v/v) FA, and applied to an ACQUITY UPLC System coupled to a Synapt G2-S mass spectrometer (Waters, Eschborn, Germany). For quantification, the sample was spiked with 10 fmol rabbit glycogen phosphorylase. Bound peptides were eluted in a linear gradient from 7 to 35% mobile Phase B (0.1% (w/v) FA in acetonitrile) for 120 min. MS acquisition was set to 50–5000 Da. Data analysis was carried out by ProteinLynx Global Server (PLGS 3.0.1, Apex3D algorithm v. 2.128.5.0, 64 bit, Waters, Eschborn, Germany) with automated determination of chromatographic peak width as well as MS TOF resolution. Lock mass value for charge state 2 was defined as 785.8426 Da/e and the lock mass window was set to 0.25 Da. Low/high energy threshold was set to 180/15 counts, respectively. Databank search query (PLGS workflow) was carried out as follows: Peptide and fragment tolerances were set to automatic, two fragment ion matches per peptide, five fragment ions for protein identification, and two peptides per protein. Primary digest reagent was trypsin with one missed cleavage allowed. The false discovery rate (FDR) was set to 4% at the protein level. MSE data were searched against the modified A. thaliana database (TAIR10, ftp://ftp.arabidopsis.org) containing common contaminants such as keratin (ftp://ftp.thegpm.org/fasta/cRAP/crap.fasta). Quantification was performed based on the intensity of the three most abundant proteotypic peptides and the comparison to the intensity of the spiked reference[18]. All quantitative proteomics data were obtained with three independent biological replicates that were measured in three technical replicates each, giving rise to nine measurements per sample.

**N-terminal peptide identification by ChaFRADIC.** Identification of N-terminal peptides by ChaFRADIC was performed according to a protocol reported by Shema and colleagues[26]. Here, one iTRAQ label (20 μl of label from iTRAQ-8plex labeling kit (AB Sciex) + 80 μl of isopropanol) was added to each sample comprising 100 μg protein. The 121, 119, and 118 labels were used for the *rpn8a ppi2* double mutant while 117, 116, and 113 were used for ppi2. After labeling, protein pellets were digested with Trypsin (ArgC-specific) and peptides were dried under vacuum before being resolubilized in 52 μl of SCX buffer A (10 mM KH2PO4, pH 2.7, 20% (v/v) ACN). For the first SCX step, 50 μl (200 μg) of each sample were on a U3000 HPLC system (Thermo Scientific), a 150 × 1 mm POLYSULFOETHYL A column (PolyLC, Columbia, US, 5 μm particle size, 200 Å pore size), and a tertiary buffer system containing SCX buffer A (10 mM KH2PO4, pH 2.7, 20% ACN), SCX buffer B (10 mM KH2PO4, 250 mM KCl, pH 2.7, 20% ACN), and SCX buffer C (10 mM KH2PO4, 600 mM NaCl, pH 2.7, 20% ACN). An optimized gradient was used to efficiently separate the peptides according to their charge states combined with automated fraction collection. Fractions corresponding to charge states +1 to +4 were collected, dried under vacuum and resuspended in 50 mM Na2HPO4, pH 7.8. Free N-termini of internal peptides were blocked by trideutero (d3)-acetylation to induce a charge state shift. Therefore, trideutero N-hydroxysuccinimide acetate (d3-NHS acetate, synthesized as described previously[36]) was added to a final concentration of 20 mM and the sample incubated for 1 h at 37 °C, followed by a second step using 10 mM d3-NHS acetate. After 2 h the reaction was quenched by addition of 60 mM glycine for 10 min, followed by addition of 130 mM hydroxylamine for 10 min, both at RT. Samples were desalted as described above and dried under vacuum. After resolubilization, each fraction was separately subjected to re-chromatography under the same conditions as above. Collected fractions were completely dried under vacuum and desalted by C18 solid phase extraction. Dried eluates were resuspended in 45 μl of 0.1% (w/v) TFA and 1/3rd of each fraction applied was analyzed by nano-LC-MS/MS using a Orbitrap Fusion Lumos mass spectrometer online-coupled to a nano RSLC HPLC system (both Thermo Scientific). LC separation was performed as described before[25]. Samples were measured in data-dependent acquisition mode using the top speed option (3 s). Survey scans were acquired from 350–1550 *m/z* at resolution of 120,000 using AGC target value of 2e5 and a maximum injection time of 50 ms. Per precursor, two MS/MS were acquired, precursors were selected using the quadrupole with an isolation width of 0.8 *m/z*. The first MS/MS (peptide identification) was acquired in the ion trap in rapid mode, with an AGC target value of 2e3, a maximum injection time of 300 ms and a normalized HCD collision energy of 35 %. The second MS/MS (reporter ion quantification) was acquired in the Orbitrap at a resolution of 15,000, with an AGC target value of 1e5, and maximum injection time of 100 ms, and a normalized HCD collision energy of 60 %. For data analysis, Proteome Discoverer software version 1.4 (Thermo Scientific) was used with Mascot 2.4 (Matrix Science) as search engine, reporter ion quantifier and percolator nodes. To enable the quantification of both classes of N-terminal peptides, those with N-terminal iTRAQ label and those with endogenous N-acetylation, we performed a two-step search strategy: First, data were searched with iTRAQ-8plex (+304.2053 Da) as fixed modification at Lys and peptide N-termini; second iTRAQ 8-plex of Lys and N-terminal acetylation (+42.0105 Da) of N-termini were set as fixed modifications.

Mass tolerances were set to 10 ppm for MS and 0.01 Da for MS/MS. Identified peptides were filtered for high confidence corresponding to an FDR ≤ 1% at the PSM level, and a search engine rank of 1. The reporter ion quantifier node was adjusted according to the search settings. Reporter areas were normalized based on a single-shot nano-LC-MS/MS analysis of 1 μg of the digested sample, without enrichment. Normalized reporter areas were used to calculate ratios and *p*-values.

**Reporting summary**. Further information on research design is available in the Nature Research Reporting Summary linked to this article.

## Data availability
All MS data were uploaded to PRIDE (https://www.ebi.ac.uk/pride) and are accessible via the identifier PXD014531 (single and double mutants), PXD014560 (MG132 dataset) and PXD017126 (ChaFRADIC dataset). A sample key for the uploaded data files with explanation is available in Supplementary Data 7. All other data are available in the text or in the supplement and in the source data file. All mutants and double mutants generate within the course of this study are available upon request to the corresponding author.

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

## Acknowledgements

We are grateful for financial support from the DFG grant BA 1902/3–2 and the European Regional Development Fund of the European Commission grant W21004490 via Land Sachsen-Anhalt to S.B. S.B. gratefully acknowledges DFG support for the acquisition of a Synapt G2-S mass spectrometer (INST 271/283–1 FUGG).

## Author contributions

J.G.—formal analysis, investigation, resources, validation, visualization data curation; S.H., D.D., G.H., G.S., R.P.Z.—data curation, formal analysis, investigation, methodology, validation; S.B.—conceptualization, formal analysis, investigation, supervision, writing.

## Competing interests

The authors declare no competing interests.
