## [Peer Review File · Nature Communications]

Reviewers' comments:

Reviewer #1 (Remarks to the Author):

This manuscript by Grimmer et al. mainly focused the effect of 26S proteasome mutants in photosynthesis, particularly in relation to chloroplast protein import deficiency. The work is extensive and impressive, generating numerous Arabidopsis mutants and performing protein quantitation with proteome analysis. The main essence of this paper is, if I understand correctly, the recovery of pale phenotype in *ppi2* when combined with 26S proteasome mutations (*rpn8a* in Figure 1; *rpa2b* and *pad1* in Figure 4). Moreover, the authors showed that *rpn8a* improves photosynthesis performance (Fig. 2). Given the previous observation that similar effects were seen in *ppi1/sp1* mutants and that SP1 is E3 ligase specifically associated with Toc components, the authors propose that general proteasomal degradation of precursor proteins in the cytosol, rather than the specific degradation of protein import machineries, controls chloroplast biogenesis. While I thought the work is deep and interesting, the manuscript was difficult to follow. It perhaps comes from several inconsistencies in the data presented in the figures. Some of those are highlighted below.

1) As mentioned above, suppression phenotype of *rpn8a*, *rpn8b*, and *pad1* (but not *sp1*) is split in Figure 1 and Figure 4A. I think it should be comparably shown in the same figure with proper controls. Also, not only pigment content but also photographs of these mutants need to be shown at the several different developmental stages.

2) In Figure 1, *ppi2* is shown as a kind of control against the double mutants, but the actual WT is better to be included (perhaps *sp1*, too).

3) Figures 1B and 1C: protein abundance is estimated based on total protein amount? If so, the change reflects the degree of greening?

4) Figure 2: I understand the result well. However, readers may like to have other lines included in this figure, such as *sp1*, *rpn8b* and *pad1*, to assess how this finding can be generalized.

5) Figure 3: Why the double mutant was not subjected to MG132 treatment in panel A? In panel B, the difference of protease activity between *ppi2* and *rpn82/ppi2* is significant (statistical analysis is lacking here)?

Although the authors attempted to compare *rpn8* mutants to *sp1*, which I understand well, they may want to pay attention to the following publication, in which ubiquitylation of chloroplast protease is reported. This suggests a possibility that a defect in cytosolic protein degradation has a pleiotropic effect in chloroplastic protein degradation.

Shen G, Adam Z, Zhang H. (2007) The E3 ligase AtCHIP ubiquitylates FtsH1, a component of the chloroplast FtsH protease, and affects protein degradation in chloroplasts. *Plant Journal*

Reviewer #2 (Remarks to the Author):

Comments on the manuscript NCOMMS-19-26321

In their study "Mild proteasomal stress improves photosynthetic performance in Arabidopsis chloroplasts", Julia Grimmer and her colleagues suggest that a slower degradation of functional protein precursors in the cytoplasm, which is caused by mild impairment of the proteasome function, would allow fine-tuned systemic regulation of organellar biogenesis and functions.

The study has some fascinating parts. The *rpn8axppi2* mutant, which the authors present in their study, is really intriguing. The loss of the *rpn8a* function in the environment of the *ppi2* mutant has an amazing effect on restoring a part of the viability of the plant cell. The underlying mechanisms would definitely be interesting for both specialists in the field and a wider readership.

Understanding double mutants is not easy because the mutations may have indirect effects that are not obvious. As for the *rpn8axppi2* mutant, there would be a relatively easy model that could serve as an attempt for an explanation. Loss of the lid-subunit Rpn8a of the 26 S proteasome decreases the turnover of cytosolic precursors, which increases the pressure on the import system of the chloroplasts. That may allow more precursors to be imported into the chloroplasts of the *rpn8axppi2* mutant than into those of the *ppi2* mutant. The chloroplast can synthesize some thylakoids and the viability of the cells in the *rpn8axppi2* mutant increases in comparison to the *ppi2* mutant. That explanation would be easier to understand than a systemic regulatory effect through proteasome activity. The authors identify 18 intact cytosolic plastid precursors that are 1.2 time higher expressed in the *rpn8axppi2* mutant than in the *ppi2* mutant, which is consistent with this explanation. That model would also work as a possible explanation for the observations for the *pad1xppi2* double mutant. The conclusion that mild impairment of the proteasome function, would allow fine-tuned systemic regulation of organellar biogenesis and functions is not entirely obvious to me.

Another point that puzzles me is the experiment with the proteasome inhibitor MG132. The inhibitor MG132 targets the 20S core of the 26S proteasome (reviewed by Momose and Watanabe, 2017), while loss of the Rpn8a subunit affects the lid. It is not obvious to me that the cell stress that is caused by MG132 in the wildtype and the *ppi2* mutant resembles the situation in the cells of the *rpn8a* and *rpn8axppi2* mutants. The authors observe that both the *rpn8a* and *rpn8axppi2* mutants and the MG132 wildtype and *ppi2* mutant have higher amounts of the 20S core, but they might have it for different reasons and assemble their proteasomes in different ways. The protein quantification by proteomics cannot give any detailed information about it, and therefore it is hard to tell what is really going on in the *rpn8a* and *rpn8axppi2* mutants.

Changing subjects, the text in the manuscript includes some jumps, and it might not be easy to read for readers who have not followed the field in detail for some time. For instance, it is not obvious what inspired the authors to study the loss of *rpn8a* function in the wildtype and the *ppi2* mutant of Arabidopsis. The *rpn8axppi2* mutant and the *rpn8a* mutant appears in the manuscript a bit like a rabbit that jumps out of the hat of a magician. Studies on the function of the Rpn11-Rpn8 heterodimer such as, for instance, Pathare et al. (2014) and Worden et al. (2014) might be worthwhile to mention? In general, if the authors would like to reach out to a wider readership, I would suggest to consider to make the manuscript more reader friendly and to give the presentation of the ideas behind the design of the experiments a clearer structure. On a side note, reference 15 is cited in a way that can be misleading.

Switching topics and moving to some details of the experimental data, the Western blots shown in figure 1B do not look entirely excellent. The bands of Lhcb4.1 in the 50 and 100 microgram samples of the *rpn8axppi2* mutant indicate transfer issues and some other bands show streaking. Are these Western blots really suitable for quantitation? As for figure 1 A and 2A, please explain the meaning of the stars. That applies also to all following figures. In addition, how is the rate of green plants in figure 2B defined and what do the light gray bars mean?

As for figure 2C, I suggest a better presentation of the PAM-measurements. It is much squeezed together and it is difficult to see the zero values for the PSII quantum yields. Please, also explain what the red crosses mean. I also suggest to add the diagram for the electron transfer rate, which would not require any new experiments, as it can be extracted for the Pam-measurements that the authors already have performed.

The datasets PXD014531 and PXD014560 would benefit from a better description of the datasets. One important point is that it is not always obvious from the names of the raw data files which samples were analyzed. The _Header files do not include this information either. Please add a sample key to the raw data files, which is not difficult to do. In addition, I suggest to include all relevant experimental details, which are needed to understand the proteomics experiments, in the description of the datasets PXD014531 and PXD014560 that it is possible to understand and use these datasets independently of the manuscript (FAIR data). In addition, I would like to ask why the authors did not submit the dataset of the ChaFRADIC experiment to ProteomeXchange? It seems that it would be worthwhile to do it.

References

Momose I, Watanabe T. Tyropeptins, proteasome inhibitors produced by *Kitasatospora* sp. MK993-dF2. *J Antibiot (Tokyo)*. 2017 May;70(5):542-550. doi:10.1038/ja.2017.9.

Pathare GR, Nagy I, Śledź P, Anderson DJ, Zhou HJ, Pardon E, Steyaert J, Förster F, Bracher A, Baumeister W. Crystal structure of the proteasomal deubiquitylation module Rpn8-Rpn11. *Proc Natl Acad Sci U S A*. 2014 Feb 25;111(8):2984-9. doi: 10.1073/pnas.1400546111.

Worden EJ, Padovani C, Martin A. Structure of the Rpn11-Rpn8 dimer reveals mechanisms of substrate deubiquitination during proteasomal degradation. *Nat Struct Mol Biol*. 2014 Mar;21(3):220-7. doi: 10.1038/nsmb.2771.

Reviewer 1:

1) As mentioned above, suppression phenotype of *rpn8a*, *rpn8b*, and *pad1* (but not *sp1*) is split in Figure 1 and Figure 4A. I think it should be comparably shown in the same figure with proper controls. Also, not only pigment content but also photographs of these mutants need to be shown at the several different developmental stages.

We agree with this reviewer and have substantially reorganized the manuscript to generate a better text flow. We now present the phenotypic characterization of all mutants in Figure 1 and provide a better explanation for the selection of the mutants and the incentive to cross them with *ppi2*. In our opinion, this improves the readability of the manuscript because the inherent logic is better to follow.

2) In Figure 1, *ppi2* is shown as a kind of control against the double mutants, but the actual WT is better to be included (perhaps *sp1*, too).

We have not shown the wildtype and *ppi2* data in the same graph because the chlorophyll concentrations span a large range, with wildtype concentrations being still 10-fold above those reached with the suppressor (20-fold higher in case of *ppi2*). The relative chlorophyll amount is now shown in one graph (Fig. 1C) and the absolute pigment amounts are provided in the source data file.

3) Figures 1B and 1C: protein abundance is estimated based on total protein amount? If so, the change reflects the degree of greening?

Yes, protein abundance was normalized to total protein amount. Normalization works best in case the specific “changes” in protein abundance in certain proteins or protein groups are buffered by a large number of proteins that remain unchanged. Therefore, the fact that photosynthetic proteins accumulate to higher abundance in the double mutant can be considered specific. But of course, their accumulation also reflects the degree of greening and vice versa, these two processes cannot be separated.

4) Figure 2: I understand the result well. However, readers may like to have other lines included in this figure, such as *sp1*, *rpn8b* and *pad1*, to assess how this finding can be generalized.

We reorganized the manuscript to present the mutants and their phenotypes in one figure (Figure 1). Figure 2 now presents a new electron microscopic (TEM) characterization of the *pad1xppi2* lines that showed significantly increased pigment concentrations compared to *ppi2* (Fig. 1). Our reasoning was that the TEM pictures would provide a better assessment of the situation of individual cells compared to a protein profiling that averages all cells independent of their status. In fact, we observed a heterogeneous thylakoid structure distribution in the *pad1xppi2* lines, with some plastids having no stacks at all while others showed up to four thylakoid stacks and a relatively well developed thylakoid membrane system (see Supplemental Fig. 2). This is consistent with an intermediate phenotype of *pad1xppi2* between *ppi2* and *rpn8axppi2*, consistent with the quantitative pigment levels. Thus, we considered that the *pad1xppi2* lines need further experimental characterization that would go beyond the scope of this manuscript here. We therefore decided to move ahead with the characterization of the better suppressor, i.e.

***rpn8axppi2* and carry the *sp1xppi2* through all the analyses as a “functional” reference in which the suppression mechanism on *ppi1* is known.**

5) Figure 3: Why the double mutant was not subjected to MG132 treatment in panel A? In panel B, the difference of protease activity between *ppi2* and *rpn82/ppi2* is significant (statistical analysis is lacking here)?

We intended to mimic proteasomal impairment/inhibition by MG132 treatment to compare it with the proteome adaptations of the double mutant. Treating the double mutant with MG132 addresses- in our opinion- a different question that is indeed worthwhile investigating, but since we already have such a large proteomics data pool in the manuscript here, we refrained from performing this analysis at this time. The data we present here show that a full inhibition of the proteasome does not mimic the effect of the mutations. This is because the proteasome is such an important pleiotropic machinery that full inhibition will undoubtedly affect other cellular processes making an interpretation of the data difficult. On the other hand, application of MG132 for a short time does not allow mimicking the effect of constantly altered proteasome activity, while short-time application in protoplasts assays does affect precursor and mature protein accumulation (Fig. 6 and Supplemental Fig. 6). We have now improved the description of the data and the motivation for the experiments with MG132. We also provide now the reasons why we think the inhibition did not mimic the effect of the mutations (see also Reviewer 2).

*Although the authors attempted to compare *rpn8* mutants to *sp1*, which I understand well, they may want to pay attention to the following publication, in which ubiquitylation of chloroplast protease is reported. This suggests a possibility that a defect in cytosolic protein degradation has a pleiotropic effect in chloroplastic protein degradation.*

We agree with this reviewer. Although we were aware of this paper, we did not include it here since a thorough analysis of all proteases in the proteomics datasets did not show any signs of an altered protease network in the mutants. Notably FtsH5 levels are significantly increased in both *sp1xppi2* and in *rpn8axppi2* double mutants (both by a factor of 1.8 and a p-value of 0.03 (*sp1xppi2*) and 0.01 (*rpn8axppi2*), but no other protease is significantly different between the double mutants and the *ppi2* single mutants. Therefore, an indirect effect of changes in the chloroplast protease network on the phenotype reported here is unlikely. We included a short discussion of this paper and the data on the proteases in the revised version of our manuscript.

Reviewer 2

*The study has some fascinating parts. The *rpn8axppi2* mutant, which the authors present in their study, is really intriguing. The loss of the *rpn8a* function in the environment of the *ppi2* mutant has an amazing effect on restoring a part of the viability of the plant cell. The underlying mechanisms would definitely be interesting for both specialists in the field and a wider readership.*

*Understanding double mutants is not easy because the mutations may have indirect effects that are not obvious. As for the *rpn8axppi2* mutant, there would be a relatively easy model that could serve as an attempt for an explanation. Loss of the lid-subunit Rpn8a of the 26 S proteasome decreases the turnover of cytosolic precursors, which increases the pressure on the import system of the chloroplasts. That may allow more precursors to be imported into the chloroplasts of the *rpn8axppi2**

mutant than into those of the ppi2 mutant. The chloroplast can synthesize some thylakoids and the viability of the cells in the rpn8axppi2 mutant increases in comparison to the ppi2 mutant. That explanation would be easier to understand than a systemic regulatory effect through proteasome activity. The authors identify 18 intact cytosolic plastid precursors that are 1.2 time higher expressed in the rpn8axppi2 mutant than in the ppi2 mutant, which is consistent with this explanation. That model would also work as a possible explanation for the observations for the pad1xppi2 double mutant. The conclusion that mild impairment of the proteasome function, would allow fine-tuned systemic regulation of organellar biogenesis and functions is not entirely obvious to me.

We agree with this reviewer and his/her explanation of the phenotype. This is exactly what we had in mind, when interpreting the data, but the original manuscript did not clearly point this out. We have now substantially revised the text, highlighting this explanation much clearer now. We also improved the logic flow of the text much better since we substantially reorganized the manuscript and the figures therein (see also below). We also deleted the term “systemic” since it is misleading in the context of the phenotypic interpretation.

Another point that puzzles me is the experiment with the proteasome inhibitor MG132. The inhibitor MG132 targets the 20S core of the 26S proteasome (reviewed by Momose and Watanabe, 2017), while loss of the Rpn8a subunit affects the lid. It is not obvious to me that the cell stress that is caused by MG132 in the wildtype and the ppi2 mutant resembles the situation in the cells of the rpn8a and rpn8axppi2 mutants. The authors observe that both the rpn8a and rpn8axppi2 mutants and the MG132 wildtype and ppi2 mutant have higher amounts of the 20S core, but they might have it for different reasons and assemble their proteasomes in different ways. The protein quantification by proteomics cannot give any detailed information about it, and therefore it is hard to tell what is really going on in the rpn8a and rpn8axppi2 mutants.

Again, point taken. We intended to see if we can increase the accumulation of photosynthetic proteins by inhibiting the proteasome activity for a certain time span. This was not successful for reasons outlined by this reviewer. We now clearly state that mutations in the lid and proteasome inhibition have distinct effects on the accumulation of photosynthetic protein, which is not surprising because of 20S proteasome activity (that is inhibited by MG132 while being unaffected in the rpn8a mutants); an explanation was added to the results part. We also separated the induction of genes for the proteasomal stress regulon at the transcriptional level from the MG132 proteomics experiments. Together this part of the manuscript has substantially improved in terms of readability.

Changing subjects, the text in the manuscript includes some jumps, and it might not be easy to read for readers who have not followed the field in detail for some time. For instance, it is not obvious what inspired the authors to study the loss of rpn8a function in the wildtype and the ppi2 mutant of Arabidopsis. The rpn8axppi2 mutant and the rpn8a mutant appears in the manuscript a bit like a rabbit that jumps out of the hat of a magician. Studies on the function of the Rpn11-Rpn8 heterodimer such as, for instance, Pathare et al. (2014) and Worden et al. (2014) might be worthwhile to mention? In general, if the authors would like to reach out to a wider readership, I would suggest to consider to make the manuscript more reader friendly and to give the presentation of the ideas behind the design of the experiments a clearer structure. On a side note, reference 15 is cited in a way that can be misleading.

The inspiration for the analysis of double mutants between proteasomal subunits and protein import mutants is now better elaborated. Most importantly, we previously observed precursor protein accumulation in *ppi2* and since these are cleared by the UPS, we intended to test for an effect of proteasomal impairment; all in light of the suppression of *ppi1* by Sp1 (Ling et al., 2012). We have furthermore added a sentence on the function of the Rpn11-Rpn8 heterodimer to the text where the *rpn8a* mutant is introduced.

Switching topics and moving to some details of the experimental data, the Western blots shown in figure 1B do not look entirely excellent. The bands of Lhcb4.1 in the 50 and 100 microgram samples of the rpn8axppi2 mutant indicate transfer issues and some other bands show streaking. Are these Western blots really suitable for quantitation? As for figure 1 A and 2A, please explain the meaning of the stars. That applies also to all following figures. In addition, how is the rate of green plants in figure 2B defined and what do the light gray bars mean?

The data of the Western Blots and the quantitative proteome analyses indicate the same abundance distribution for the TOC components and for photosynthetic proteins in the tested genotypes, we therefore think that the Western quantification is reliable. For better transparency, all the Western Blots used for quantification were deposited in the source data files. We have now moved the stress resilience and the *rpn8* growth data to the supplement (formerly figure 2). “Rate of green” plants indicates those plants that were already green after the indicated time span, as judged by eye. The light gray bars refer to the germination rate (% germinated seeds) to assess any potential bias here.

As for figure 2C, I suggest a better presentation of the PAM-measurements. It is much squeezed together and it is difficult to see the zero values for the PSII quantum yields. Please, also explain what the red crosses mean. I also suggest to add the diagram for the electron transfer rate, which would not require any new experiments, as it can be extracted for the Pam-measurements that the authors already have performed.

The PAM measurements are now collectively presented in one figure (Fig. 3), the graphs are larger, their readability is improved and the ETR is provided.

The datasets PXD014531 and PXD014560 would benefit from a better description of the datasets. One important point is that it is not always obvious from the names of the raw data files which samples were analyzed. The _Header files do not include this information either. Please add a sample key to the raw data files, which is not difficult to do. In addition, I suggest to include all relevant experimental details, which are needed to understand the proteomics experiments, in the description of the datasets PXD014531 and PXD014560 that it is possible to understand and use these datasets independently of the manuscript (FAIR data). In addition, I would like to ask why the authors did not submit the dataset of the ChaFRADIC experiment to ProteomeXchange? It seems that it would be worthwhile to do it.

We have added a file to the manuscript that serves as a data key (Supplementary Table 7). We were unable to submit an Excel File directly to the original PRIDE upload, as this was rejected by the database. We also uploaded the CHAFRADIC data to PRIDE, that are now available under PXD017126.

REVIEWERS' COMMENTS:

Reviewer #1 (Remarks to the Author):

Inclusion of the images from single and double mutants in Figure 1 makes the manuscript easier to follow, and at least we assess how the double mutants look. However, my honest impression is that the suppression of growth in *rpn8a/ppi2* is, when compared to *ppi2*, is VERY weak. Then the authors decided to show the ratio of pigments/fresh weight instead of pigment content - this is fine to demonstrate the significant difference between the single and double mutants, but I am not fully satisfied because the substantial difference between WT and the mutants is neglected. At least, the authors should mention explicitly somewhere in the text that Chl content of *ppi2 rpn8a/ppi2* is 10-fold lower than WT, thus the suppression of *ppi2* phenotype is only minor when compared to WT.

In this context, I did not agree with the statement in Discussion (316-317) 'Mutations in the lid subunit Rpn8a and the core subunit PAD1 partially complement the *ppi2* mutant phenotype'. Perhaps the phenotype observed in the double mutants is not to the extent that we can state it as partial complementation.

Figure 6. I was puzzled when I read the legend pointing GFP antibodies here, because the fact that the precursor proteins were GFP fusions was not mentioned in the text or the figure at all. This panel therefore needs some revision, like putting caption like what are seen in Figure S6.

Reviewer #2 (Remarks to the Author):

Comments on the manuscript

The revised version of this manuscript has a much improved text and presentation of the results. The study is fascinating and of interest for both specialist in the field and a broad readership. Nevertheless, there are some point, which I would like the authors to consider for a revision.

Until figure 5, the text is excellent and a reader can follow it without difficulties. I suggest to move the the section about the elevated precursor level in the cytosol of the double mutants in the end of the manuscript. It would perhaps better fit into the manuscript at a place between figure 4 and 5? There is also a minor question about figure 1 C. Has the y-axis the same scale for *ppi2* and the double mutants as for the wildtype and the other single mutants? It appears that the y-axis for *ppi2* and the double mutants is missing.

Figure 5 is not easy to understand. I wonder, if the authors possibly could consider more the different mechanisms of function that are known for the 26S and the 20S proteasome, when they present these results?

The *rpn8a* mutant has no lid and it shows no phenotype under the controlled growth conditions of this study. This observation fits together with the observation by Paci et al. (2016) that a region close to the C-terminus of yeast Rpn8 is needed for assembling the intact 26S proteasome. As the *rpn8a* mutant cannot perform de-ubiquitination, it relies on 20S activity. As reviewed by Ben-Nissan (2014), 20S proteasome can recognize and degrade proteins that contain unstructured regions because of oxidation, mutation or aging. Apparently, the proteolytic capacity of 20S in the *rpn8a* mutant is sufficient to ensure that this mutant can live with 20S without turning into a phenotype. The ability of the proteasome to function without the lid can in part also explain the data for the *ppi2xrpn8a* double mutant, although this mutant has a lid and would be expected to also have the ubiquitination-controlled pathway of 26S proteasome function. Taking this into

account, the data for the *rpn8a* mutants in figure 5 would not simply suggest a compensatory upregulation of the 20S proteasome but also a shift to different pathways for proteasome function.

The authors use the MG132 proteasome inhibitor to mimic the effect of *rpn8a* mutation on photosynthetic protein abundance. Is this convincing? The data that the authors present show that the MG132-treatment is accompanied by a significant increase of the amount of 20S, which in turn may affect the expression of the lid and base subunits. Mechanistically, this would be different from mimicking loss of *Rpn8a*. The authors suggest that loss of *Rpn8a* and inhibition of 20S by MG132 have different effects on the import of chloroplast proteins in wildtype plants and in plants with *ppi2* background. That is reasonable, but the line of reasoning is not entirely clear and the authors might consider clarifying it and try to achieve a better separation between presenting and discussing their results in this part of the manuscript (line 251 and following).

To add a brief technical comment, I understand that space for presenting figures is limited, but perhaps there is solution to present the data for the lid and the base with the same scale for the y-axis?

References

Ben-Nissan G, Sharon M. Regulating the 20S proteasome ubiquitin-independent degradation pathway. *Biomolecules*. 4:862-884, 2014. doi: 10.3390/biom4030862.

Paci A, Liu PX, Zhang L, Zhao R. The proteasome subunit *Rpn8* interacts with the small nucleolar RNA protein (snoRNP) assembly protein *Pih1* and mediates its ubiquitin-independent degradation in *Saccharomyces cerevisiae*. *J Biol Chem*. 291:11761-11775, 2016. doi: 10.1074/jbc.M115.702043.

Comments on supplementary material and the data submitted to EBI-Pride

The document 219310_1_supp_4359005_q494t2.pdf is a valuable supplement and may well be added to the manuscript, but it would help the reader if the figures would have a description that integrates them better into the context of the manuscript.

Please consider some additions and corrections to the data that you submitted to EBI-Pride. You can only make changes to the data that you submitted before they are published. When your data have received a DOI, they are locked and the Pride team will not be able to add corrections or revisions.

Private Project PXD014560

Please give the full reference of Helm et al. 2014. Please avoid empty key words as far as possible ("unknown") and add software and experiment type. Some characters were converted to unreadable symbols. Please replace them by simple text.

Private Project PXD014531

Please add a csv file with descriptions of the raw data file like in your project PXD014560. Please give the full reference of the publication of Helm et al. (2014). The description of the data in a repository should be independent from an article, for which these data were used and the metadata should be rich. Like in project PXD014560, please avoid empty key words and add acquisition software and experiment type.

Private Project PXD017126

Please correct undefined characters such as \hat{A} and remove them or replace them by simple text. Please give the full reference of the publication of Shema et al. Like in the previous projects please avoid empty key words and add the data acquisition software.

As for the Excel file *ChaFRADI_JG.xlsx*, you may keep it in this dataset, but please provide the

sheets of this file also as single csv files and add a readme file that describes their content. The reason for that is that CSV is a recommended data format for archiving, which XLSX is not. Please, have also a look at the sheet of table 3 (Tabelle 3) in the file ChaFRADI_JG.xlsx. Was it supposed to be empty?

Response to reviewer comments

Reviewer #1:

*At least, the authors should mention explicitly somewhere in the text that Chl content of *ppi2 rpn8a/ppi2* is 10-fold lower than WT, thus the suppression of *ppi2* phenotype is only minor when compared to WT.*

*In this context, I did not agree with the statement in Discussion (316-317) ‘Mutations in the lid subunit *Rpn8a* and the core subunit *PAD1* partially complement the *ppi2* mutant phenotype’. Perhaps the phenotype observed in the double mutants is not to the extent that we can state it as partial complementation.*

We agree to these point brought up be the reviewer and we modified the text to comply with this critique. First, we have added the statement that the double mutants achieved a doubling of the chlorophyll content of *ppi2* while still remaining 12-fold lower than wildtype levels. The data are provided in the source data file. And second, we have replaced all general references to “suppression effects” by more precisely mentioning what exactly is suppressed, e.g. pigment content is restored to higher levels, photosynthetic activity in increased, elevated thylakoid stacking is observed and so on. By this, we avoid the impression of a full complementation effect. We also provide an explanation why the double mutant is still limited in its growth, i.e. scavenging of extra excitation energy is necessary most likely because the *ppi2* mutant has pleiotropic defects that impede efficient use of photosynthetic electrons (as indicated by the NPQ measurements with the double mutant compared to the single mutant).

Figure 6. I was puzzled when I read the legend pointing GFP antibodies here, because the fact that the precursor proteins were GFP fusions was not mentioned in the text or the figure at all. This panel therefore needs some revision, like putting caption like what are seen in Figure S6.

We have added this information to the figure caption and the text, and also added a panel to the figure itself.

Reviewer #2:

Until figure 5, the text is excellent and a reader can follow it without difficulties. I suggest to move the the section about the elevated precursor level in the cytosol of the double mutants in the end of the manuscript. It would perhaps better fit into the manuscript at a place between figure 4 and 5?

We have considered this suggestion thoroughly and came to conclude that we would like to leave the order of figures as it is. The reason is that it is more logical for the reader – in our opinion from the plant scientists’ perspective – to have the effect on precursor levels at the end of the manuscript. First, we investigated what changes occur concerning proteasome composition and activity, and then investigate the effect on chloroplast proteins with this knowledge. We agree that it would work the other way round, but we think this would emphasize a different aspect of the MS. Thus, we refrained from modifying the text flow.

There is also a minor question about figure 1 C. Has the y-axis the same scale for ppi2 and the double mutants as for the wildtype and the other single mutants? It appears that the y-axis for ppi2 and the double mutants is missing.

Yes, they are the same scale, the scale is % from the reference point (100%).

Figure 5 is not easy to understand. I wonder, if the authors possibly could consider more the different mechanisms of function that are known for the 26S and the 20S proteasome, when they present these results?

The rpn8a mutant has no lid and it shows no phenotype under the controlled growth conditions of this study. This observation fits together with the observation by Paci et al. (2016) that a region close to the C-terminus of yeast Rpn8 is needed for assembling the intact 26S proteasome. As the rpn8a mutant cannot perform de-ubiquitination, it relies on 20S activity. As reviewed by Ben-Nissan (2014), 20S proteasome can recognize and degrade proteins that contain unstructured regions because of oxidation, mutation or aging. Apparently, the proteolytic capacity of 20S in the rpn8a mutant is sufficient to ensure that this mutant can live with 20S without turning into a phenotype. The ability of the proteasome to function without the lid can in part also explain the data for the ppi2xrpn8a double mutant, although this mutant has a lid and would be expected to also have the ubiquitination-controlled pathway of 26S proteasome function. Taking this into account, the data for the rpn8a mutants in figure 5 would not simply suggest a compensatory upregulation of the 20S proteasome but also a shift to different pathways for proteasome function.

We agree to the conclusion that the mutation induces a quantitative shift from 26S to 20S proteasome activity. This was already mentioned in the previous version of the manuscript; and for the revision we have added a statement to the results section together with the “Paci et al.” reference to further emphasize this point. In fact, we think that the functional significance of the 20S pathway becomes visible in the lower expression of GST as stress markers (Supplemental Fig. 4) and also in the effect of MG132 inhibition, that did not result in an increased accumulation of photosynthetic proteins.

The authors use the MG132 proteasome inhibitor to mimic the effect of rpn8a mutation on photosynthetic protein abundance. Is this convincing? The data that the authors present show that the MG132-treatment is accompanied by a significant increase of the amount of 20S, which in turn may affect the expression of the lid and base subunits. Mechanistically, this would be different from mimicking loss of Rpn8a. The authors suggest that loss of Rpn8a and inhibition of 20S by MG132 have different effects on the import of chloroplast proteins in wildtype plants and in plants with ppi2 background. That is reasonable, but the line of reasoning is not entirely clear and the authors might consider clarifying it and try to achieve a better separation between presenting and discussing their results in this part of the manuscript (line 251 and following).

We have rephrased the paragraphs to make our reasoning clearer and to avoid the impression, that we think MG132 could mimic the loss of a proteasomal subunit.

Therefore, we deleted the term “mimic” and rephrased the first sentence of the relevant paragraph to read more clearly as follows: “To test for the effect of inhibitor-induced proteasomal stress on photosynthetic protein abundance, we treated wildtype and *ppi2* plants with MG132 and compared the proteomes of inhibitor-treated plants with the proteomes of *rpn8a* single and *rpn8a ppi2* double mutants.”

We also rephrased the subsequent paragraph that should now be better readable.

To add a brief technical comment, I understand that space for presenting figures is limited, but perhaps there is solution to present the data for the lid and the base with the same scale for the y-axis?

Because of the limited space, we see no other possibility to present these data. We think that the panel-like separation of lid, base and 20S is intuitive, clearly indicating the different effects of mutation and inhibitor-treatment on the different modules of the proteasome.

The document 219310_1_supp_4359005_q494t2.pdf is a valuable supplement and may well be added to the manuscript, but it would help the reader if the figures would have a description that integrates them better into the context of the manuscript.

We agree and have added an explanatory slide to integrate the source data with the figures for which they were used. This integrates the data better with the manuscript (see also response to editor) as it is now much clearer which data are underlying which figure.

Please consider some additions and corrections to the data that you submitted to EBI-Pride. You can only make changes to the data that you submitted before they are published. When your data have received a DOI, they are locked and the Pride team will not be able to add corrections or revisions.

Private Project PXD014560

Please give the full reference of Helm et al. 2014. Please avoid empty key words as far as possible (“unknown”) and add software and experiment type. Some characters were converted to unreadable symbols. Please replace them by simple text.

Private Project PXD014531

Please add a csv file with descriptions of the raw data file like in your project PXD014560. Please give the full reference of the publication of Helm et al. (2014). The description of the data in a repository should be independent from an article, for which these data were used and the metadata should be rich. Like in project PXD014560, please avoid empty key words and add acquisition software and experiment type.

Private Project PXD017126

Please correct undefined characters such as Â and remove them or replace them by simple text. Please give the full reference of the publication of Shema et al. Like in the previous projects please avoid empty key words and add the data acquisition software.

As for the Excel file ChaFRADI_JG.xlsx, you may keep it in this dataset, but please provide the sheets of this file also as single csv files and add a readme file that describes their content. The reason for that is that CSV is a recommended data format for archiving, which XLSX is not. Please, have also a look at the sheet of table 3 (Tabelle 3) in the file ChaFRADI_JG.xlsx. Was it supposed to be empty?

We have made contact with the PRIDE support team concerning all requests highlighted above. As far as I can judge, all the uploads are now as requested and the additional csv files were uploaded and are visible. The doi for the references has been provided and the software is mentioned in the descriptions.